# Early Prediction of Physical Performance in Elite Soccer Matches—A Machine Learning Approach to Support Substitutions

**DOI:** 10.3390/e23080952

**Published:** 2021-07-25

**Authors:** Talko B. Dijkhuis, Matthias Kempe, Koen A. P. M. Lemmink

**Affiliations:** 1Department of Human Movement Sciences, University of Groningen, A. Deusinglaan 1, 9713 AV Groningen, The Netherlands; m.kempe@umcg.nl (M.K.); k.a.p.m.lemmink@umcg.nl (K.A.P.M.L.); 2Institute of Communication and ICT, Hanze University of Applied Science, Zernikeplein 11, 9747 AS Groningen, The Netherlands

**Keywords:** fatigue, decision-support, football

## Abstract

Substitution is an essential tool for a coach to influence the match. Factors like the injury of a player, required tactical changes, or underperformance of a player initiates substitutions. This study aims to predict the physical performance of individual players in an early phase of the match to provide additional information to the coach for his decision on substitutions. Tracking data of individual players, except for goalkeepers, from 302 elite soccer matches of the Dutch ‘Eredivisie’ 2018–2019 season were used to enable the prediction of the individual physical performance. The players’ physical performance is expressed in the variables distance covered, distance in speed category, and energy expenditure in power category. The individualized normalized variables were used to build machine learning models that predict whether players will achieve 100%, 95%, or 90% of their average physical performance in a match. The tree-based algorithms Random Forest and Decision Tree were applied to build the models. A simple Naïve Bayes algorithm was used as the baseline model to support the superiority of the tree-based algorithms. The machine learning technique Random Forest combined with the variable energy expenditure in the power category was the most precise. The combination of Random Forest and energy expenditure in the power category resulted in precision in predicting performance and underperformance after 15 min in a match, and the values were 0.91, 0.88, and 0.92 for the thresholds 100%, 95%, and 90%, respectively. To conclude, it is possible to predict the physical performance of individual players in an early phase of the match. These findings offer opportunities to support coaches in making more informed decisions on player substitutions in elite soccer.

## 1. Introduction

Soccer is a highly competitive and physically demanding sport. The physical demand is highlighted by an increase in ball (game) speed by 15% over the last 50 years [1]. A cohesive body of research points out that player fatigue leads to a decline in their running activities. For instance, in a team participating in the Australian national soccer league, total distance, average speed, high-intensity running distance, and very high-intensity running distance decreased significantly from the first to the second half by 7.92, 9.47, 10.10, and 10.99%, respectively [2]. In similar fashion, in the Italian A series, a team showed a significant reduction between the first and second half in high-intensity running distance (−14.9%) [3]. These examples highlight that players are unable to perform maximally throughout a match [4]. Information on this drop in performance is essential for players and coaches. A recent study showed that running performance parameters (e.g., the number of accelerations or decelerations and the distance covered in different speed categories) affect successful soccer performance for some playing positions [5]. As most soccer matches are often decided by just one goal [6], a drop in physical performance can make the difference between winning and losing. Therefore, teams and coaches need to identify players that physically underperform in a match as early as possible to adapt their style of play or substitute these players. In general, an injury of a player, necessary tactical changes, or underperformance of a player causes substitutions (for an overview, see Hills et al., 2018) [7]. Substitution may be the most powerful tool of a coach to influence the match. Substitutions can minimize or offset the effects of fatigue of the team as substitutes cover more distance and perform more high-intensity actions relative to entire-match players [8]. According to the Fédération Internationale de Football Association (FIFA) COVID-19 2020 rules, a coach has five substitution options in a match, implicating fitness of the individual player and physical performance has more impact on substitution than before COVID-19 [9].

To identify a physically underperforming player, coaches can base their decision on real-time motion data. To record and monitor real-time motion data, multi-camera position tracking systems, such as SportVU and TRACAB^®^ systems, are most commonly used in professional leagues [10]. However, one has to constantly monitor and analyze several physical variables of all eleven players. As highlighted in a survey paper by Nosek, Brownlee, Drust and Andrew (2020) [11], staff and IT solutions struggle with giving helpful feedback to the coach after training sessions due to the amount and complexity of the data and their often inconclusive communication [11]. In order to enable helpful, timely feedback, Robertson advocated using machine learning approaches as decision support for the coach [12]. Decision support provides superior efficacy when the volume of the data is large, and the data are complex [13]. An in-match physical performance prediction and decision support using machine learning is a novelty that has not yet been realized for team sports.

In order to build an in-match physical performance prediction and decision support, models have to be based on derived time-motion data variables. These derived time–motion variables can be divided into type 1 or type 2 [14]. Type 1 variables include external load measures, such as distance covered and distance covered in the speed category. Type 2 variables include load measures related to changes in velocity, such as accelerations, decelerations, and summarized variables such as metabolic power and energy expenditure. Researchers have tried to quantify physical performance decline as a decrease in various type 1 variables. It turns out that during the match, the distance covered and the distance in the speed category decrease [2,3,15]. However, type 2 variables are more sensitive to identifying in-match physical performance decline than type 1 variables [14]. Furthermore, condense variables such as metabolic power are specially equipped for identifying in-match performance decline. They hold a more linear relationship with fatigue and include accelerations and decelerations in their calculation [16]. These findings highlight the sensitivity of type 2 variables for physical performance decline. Therefore, we include both the more common type 1 and the more sensitive type 2 variables in our prediction models. Contextual factors like home or away, rank position, and score show a difference in the overall distance covered [17]. Although we acknowledge the contextual factors such as home or away, rank position and score, we excluded these contextual factors in this proof-of-concept study. Instead, we focused on the individual player in-match motion data.

The study’s goal is to predict the in-match physical performance decline of the individual soccer player using machine learning. To our knowledge, no prior study in professional soccer has investigated the in-match physical performance prediction using machine learning techniques enabling decision support for the coach on substitutes. We aim to prove: (1) if physical performance decline can be identified using both type 1 and type-2 variables; (2) if substitutes perform better than entire-match players on both type 1 and type 2 variables and (3) if the degree of physical performance of a player can be predicted in an early stage of the match using machine learning models for type 1 and type 2 variables.

## 2. Materials and Methods

### 2.1. Experimental Approach to the Problem

For our study, we retrospectively collected the in-match position tracking data from 302 competitive professional soccer matches between 18 teams during the Dutch ‘Eredivisie’ 2018–2019 season.

For our analysis, two matches with erroneous and missing data were excluded. Additionally, the extra time at the end of the first and second half and goalkeepers were excluded from the dataset.

The effect of substitution on the match was controlled by identifying both entire-match players and substitutes. Thus, entire-match players played the full match, while the substitutes entered the match at a later stage.

### 2.2. Subjects

Four hundred and eighty players participated in the 300 matches. Four thousand nine hundred and thirty-five times, entire-match players were identified. In addition, 1533 substitutes were identified.

The majority of substitutions happened at half-time (50 min mark) and between the 60 and 90 min marks (Figure 1). The number of substitutions in the first half and the 55 min mark is significantly lower (*p* < 0.001) compared to the second half and between the 60 and 90min marks.

The Ethics Committee CTc UMCG of the University Medical Center Groningen, The Netherlands, approved the study, approval number: 201800430.

### 2.3. Data

The sample includes tracking data of all players in 302 matches. The players’ time, position, speed, and acceleration were detected and recorded by the SportsVU optical tracking system (SportsVU, STATS LLC, Chicago, IL, USA). Linke et al. (2018) tested the SportsVU optical tracking system and rated the system as being adequately reliable [18].

### 2.4. Variables

The type 1 variables distance covered and distance in the speed category, and the type 2 variable energy expenditure in the power category were applied to examine the decline in physical performance [14]. The variables were calculated as (i) distance covered per five minutes, 15 min, half, and entire match [15,19,20]; (ii) distance in the speed category per five minutes, 15 min, half, and entire match. The speed categories were categorized as Very Low Intensity Running (VLIR; 0.7–7.2 km·h^−1^), Low Intensity Running (LIR; 7.2–14.4 km·h^−1^), Medium Intensity Running (MIR; 14.4–19.8 km·h^−1^), High Intensity Running (HIR; 19.8–25.1 km∙h^−1^), and Very High Intensity Running (VHIR; >25.2 km∙h^−1^) [15,21]; (iii) energy expenditure in the power category per five minutes, 15 min, half, and entire match, calculated conforming to Osgnach et al. [22]. The power categories were categorized as Low Power (LP; from 0 to 10 W·kg^−1^), Intermediate Power (IP; from 10 to 20 W·kg^−1^), High Power (HP; from 20 to 35 W·kg^−1^), Elevated Power (EP; from 35 to 55 W·kg^−1^), and Maximal Power (MP; >55 W·kg^−1^) [22]. The descriptive statistics of the variables were calculated for entire-match players and substitutes and reported as mean ± standard deviation for each variable. The difference between entire-match players and substitutes was reported for all variables as well.

### 2.5. Statistical Analysis

For the statistical analysis, we used the statsmodels package 0.11.1 in Python 3.7.2. The statistical analysis was performed for the variables distance covered, distance covered in the speed category, and energy expenditure in the power category. First, the normality of the variables was checked for entire-match players for the first half, the second half, and the 15 min periods of both halves. The normality of the variables was checked for substitutes in the second half and 15 min periods in the second half. The Kolmogorov–Smirnov test determined the normality of the variables. No normal distribution was found for both entire-match players and substitutes in the variables (i) the distance covered (*p* < 0.001), (ii) the distance covered in the speed category in all speed categories (*p* < 0.001), (iii) the energy expenditure in the power category in all power categories (*p* < 0.001). The Kruskal–Wallis test evaluated the differences between the different periods and variables. There were significant differences between every period and variable (*p* < 0.001) for both entire-match players and substitutes. As a measure of effect size, epsilon squared (*ε*^2^) was calculated for the Kruskal–Wallis test, and values from 0 to 1 indicate no relationship to a perfect relationship, respectively [23]. In the event of a significant difference, Conover post-hoc tests were used to identify any localized effects. The variable pairwise comparisons were used to reject the null hypothesis (*p* < 0.01). Statistical significance was set at *p* < 0.05.

The source code, access to the data, and corresponding Jupiter notebooks of the statistics procedure are available as open-source software on Github (https://github.com/dijkhuist/Early-Performance-Prediction-Machine-Learning-in-Soccer, accessed on 24 July 2021).

### 2.6. Machine Learning

To predict the physical performance of individual players, machine learning models were constructed for each of the variable distance covered, distance covered in the speed category, and energy expenditure in the power category. The physical performance differences between players were eliminated by individualization and normalization of the variables and outcome measures. Variables were calculated per five-minute period of the match. The performance in the current match was compared to the average individual performance of a player over the whole season. In other words, the mean value of the performance variable over the entire season based on all entire matches by an individual player was set as a personal baseline. We further calculated these baseline values for each of the 18 5 min periods of a match. Given this approach, we could calculate a relative individual performance for each player. All constructed features are presented in Table 1.

To predict the underperformance of a player during the match, the underperformance was classified as not achieving 100%, 95%, or 90% of the entire season average of the individual player. The outcome measures were distance (m) (for distance covered and distance in the speed category) and energy expenditure (kJ·kg^−1^) (for energy expenditure in the power category). The machine learning process is visualized in Figure 2. The tracking data were used to calculate physical performance variables per individual player, as described before, and labeled as underperforming or not. After that, the dataset was split into a 70% training set and a 30% test set. Subsequently, the training set was resampled to have an equal division of performing and underperforming labels using the SMOTE method [24]. Machine learning models were generated using the learning algorithms, and the test set was applied to identify the physical performance of the individual player.

Since there is no linear relationship in physical performance during the soccer match, tree-based algorithms such as the Random Forest algorithm and the Decision Tree algorithm were applied. Conducting the machine learning models was combined with parameter tuning, randomized search, and cross-validation [25]. A simple Naïve Bayes classifier was used as the baseline model to highlight the validity of the tree-based algorithms. As it is common practice for evaluating machine learning approaches, Random Forest and Decision Tree should outperform the simple Naïve Bayes baseline classifier. The following overall performance measures were calculated for each model: accuracy, precision, recall, F1-score, and area under the curve (AUC). The scikit-learn package 0.23.1 in Python 3.7.2 was used to construct and judge the machine learning models’ performance. The source code, access to the data, and corresponding Jupiter notebooks of the machine learning procedure is available as open-source software on Github (https://github.com/dijkhuist/Early-Performance-Prediction-Machine-Learning-in-Soccer, accessed on 24 July 2021).

## 3. Results

### 3.1. Physical Performance

#### 3.1.1. Entire-Match Players

In general, the physical performance of players participating in the entire match declined throughout the match. The visualization of the distance covered is represented in Figure 3. The average distance covered declined over time from 5275 ± 223 m in the first half to 4906 ± 225 m in the second half (*p* < 0.001, ε^2^ = 0.43).

The visualization of the distance in the speed category of the entire-match players can be found in Figure 4. The distance covered in the speed category showed a decline in the average distance in the speed category LIR from 2345 ± 170 m in the first half to 2092 ± 148 m in the second half (*p* < 0.001, ε^2^ = 0.43), MIR from 888 ± 87 m in the first half to 792 ± 77 m in the second half (*p* < 0.001, ε^2^ = 0.26), and HIR from 290 ± 38 m in the first half to 268 ± 35 m in the second half (*p* < 0.001, ε^2^ = 0.08). VLIR increased during the second half from 564 ± 66 m in the first 15 min to 603 ± 69 m in the last 15 min (*p* < 0.001, ε^2^ = 0.05), and the distance covered in the speed categories HIR and VHIR almost stay stable during the entire match.

The descriptives of the distance in the power category of the entire-match players are visualized in Figure 5. The energy expenditure in the power category shows a decline in the average energy expenditure in the power categories IP from 7.26 ± 1.24 kJ·kg^−1^ in the first half to 6.47 ± 1.14 kJ·kg^−1^ in the second half (*p* < 0.001, ε^2^ = 0.75), HP from 3.52 ± 0.76 kJ·kg^−1^ in the first half to 3.13 ± 0.67 kJ·kg^−1^ in the second half (*p* < 0.001, ε^2^ = 0.70), and EP from 1.47 ± 0.44 kJ·kg^−1^ in the first half to 1.32 ± 0.40 kJ·kg^−1^ the second half (*p* < 0.001, ε^2^ = 0.75), while energy expenditure in the power categories LP and MP almost stayed stable during the match.

#### 3.1.2. Entire-Match Players versus Substitutes

The average total distance covered by substitutes is higher than the average total distance covered by entire-match players: 5123 ± 397 m versus 4906 ± 225 m (*p* < 0.001, ε^2^ = 0.12) in the second half. In addition, there was a significant difference in distance covered between substitutes and the entire-match players between 60–75 min (*p* < 0.001, ε^2^ = 0.19) and 75–90 min of the match (*p* < 0.001, ε^2^ = 0.12) (Figure 6).

The distance covered in the speed category for VLIR (*p* < 0.001, ε^2^ = 0.32), LIR (*p* < 0.001, ε^2^ = 0.75), and MIR (*p* < 0.001, ε^2^ = 0.23) of the substitutes in the second half is higher than the entire-match players in the second half. Furthermore, distance covered in the speed category showed a decline for the entire-match players in the speed categories MIR, HIR, and VHIR in the second half, while there was no such decline for substitutes.

The energy expenditure of the substitutes was higher in the second 15 min period (7.11 ± 0.86 kJ·kg^−1^ vs. 6.69 ± 0.71) (*p* = 0.007, ε^2^ = 0.002) and the last 15 min period (7.31 ± 0.72 vs. 6.53 ± 0.72) (*p* < 0.001, ε^2^ = 0.02) of the second half compared to the entire-match players. In contrast to the substitutes, entire-match players showed a decline in energy expenditure over the three 15 min periods in the second half (*p* < 0.001, ε^2^ = 0.06).

### 3.2. Machine Learning

The three prediction models for the three different thresholds of 100%, 95%, and 90% of a player’s average physical match performance showed differences in accuracy and f1 scores for both tree-based and baseline models. These differences were primarily due to the reduced number of underperformers in the 90% category. While the split between over- and underperformers is 50% for the 100% thresholds, the number of underperformers decreases to 1% for the 90% thresholds (Table 2). This naturally favors the correct prediction of performers and impedes the minority category (underperformers). Random Forest and Decision Tree outperformed Naïve Bayes in precision and recall for all three variables (distance covered, distance covered in the speed category, energy expenditure in the power category) and thresholds (Table 3). Overall, the Random Forrest approach showed the best performance for all variables. Comparing the three different variables, energy expenditure in the power category showed the best score for precision in every threshold, and therefore provided the best prediction models.

Overall, the precision of classifying underperforming players was increasing during the match. After 15 min applying either Random Forest or Decision Tree distance in the speed category and energy expenditure in the power category showed a precision of respectively 0.91, 0.88, and 0.92 for the thresholds 100%, 95%, and 90%. The baseline model Naïve Bayes was less precise than Decision Tree and Random Forest (Figure 7).

## 4. Discussion

The main goal of this study was to explore the possibility of predicting physical performance of individual players and provide decision support for coaches to help them make an informed decision on player substitutions. Our study focused on a player’s physical performance within the match, making the identification of underperforming players critical points. In line with previous research, this study revealed that entire-match players show a significant decline in physical performance during the match in distance covered, distance covered in the speed category, and energy expenditure in the power category variables [4,7]. While earlier studies found a 10–15% reduction in the HIR and VHIR from the first to the second half [2,3], our results did not show any decline in these high-intensity type 1 variables. Thereby, our findings are in agreement with more recent studies [26,27]. Furthermore, our results replicate the study of Liu et al. [26], who found that time spent in the very low intensity (VLIR) category is increasing, while time in medium intensity categories is decreasing (LIR and MIR), and time in high-intensity categories are stable throughout a match. The same pattern can be seen for the energy expenditure in different power categories. Given these results, we can support our first hypothesis that type 1 and type 2 load variables can identify decreasing player performance throughout a match.

In order to answer our second research question, we found that substitutes perform better than entire-match players for both type 1 and type 2 variables. Most of the substitutions occur at half-time and during the 60–90 min period, which aligns with previous research [27]. In agreement with the literature, substitutes who had been introduced during the second half covered more distance and performed more high-intensity activities relative to entire-match players over the same period [8]. In addition, second-half substitutes spent more energy in higher power categories [28]. As substitutes demonstrate higher values in physical performance variables than the entire-match players, the substitution of underperformers may improve the team’s performance and make the difference between winning and losing [5]. This study’s machine learning models can identify and predict a player’s physical performance in an early stage of the match. The Random Forest model outperformed both the Decision Tree and Naïve Bayes algorithm. For every threshold, the Random Forest model identified the underperformers and performers best. The precision of the variable energy expenditure in the power category outperformed models based on the variables distance covered and distance covered in the speed category. The outperformance of the variable energy expenditure in the power category illustrates that the more advanced type 2 variable is most sensitive to recognizing a player’s physical performance in an early stage of the match. The stronger the relationship in reality between the variable and the outcome, the higher the precision of the machine learning model that may be expected [29]. Following these arguments, the main finding of our study is that our machine learning models could reliably identify and predict the physical performance of a player after 15 min in the match. The early prediction of physical performance can support a decision support system as advocated by Robertson [12] and further illustrates the opportunities provided by machine learning in player monitoring during the match.

A limitation of the study is the exclusion of contextual factors, such as how home or away, rank position, position system, and score show a difference in the overall distance covered [17]. Although these contextual factors on their own influence the overall distance of the team, to generate a machine learning model on individual physical performance, every combination of the contextual factors needs to be sufficiently present in the data. Not every combination of an individual player, home or away, rank position, position system, and score will be present in one season. A coach will need to use his or her insight and knowledge to judge the prediction of physical underperformance on its merits. The use of a machine learning approach also goes in hand with some limitations. To conduct a reliable model for an individual player, there must be enough entire-match data available. We did not identify any literature in soccer to refer to the amount of data needing to be available. In the literature on fitness trackers, it is found that three days of repeated measures is necessary to represent adults’ normal activity levels with an 80% confidence [30]. In parallel, three entire matches for a player may be sufficient to identify their average physical performance. A method to conduct a reliable model is to retrain models frequently and monitor precision to identify the optimum amount of data [31].

Another limitation is that physical underperformance is just one of several reasons for a coach to substitute a player. Substitutions can also be initiated by a player’s injury, necessary tactical changes (e.g., because of being behind in a match), or tactical underperformance of a player [7]. In our study, the data was limited to the individual player’s speed, acceleration, and distance measures. Next to contextual influences [17], other physiological markers of fatigue, such as individual measures including heart rate, breathing, and body temperature, were not included. Including contextual influences and physiological markers of fatigue in the machine learning model could enable a more informative system. Finally, the thresholds of physical underperformance were randomly chosen, and the 90% threshold is relatively rarely seen.

## 5. Conclusions

Our study confirmed that the identification of physical performance could be based on type 1 and type 2 variables calculated from the position tracking systems. Additionally, substitutes perform better than entire-match players in both type 1 and type 2 variables. The appliance of machine learning enables the prediction of a player’s physical performance in an early stage in the match whereby the more sensitive type 2 variable outperforms the type 1 variables in the precision of the prediction.

### Practical Implications

These findings show that it is possible to identify underperforming players in an early stage in the match. Applying machine learning in combination with monitoring the energy expenditure in the power category during the match enables real-time support for the coach to decide on substitutions. As the nature of the game is the same for many leagues, monitoring expenditure in the power category can be of use in many other environments than Dutch elite soccer. A precondition for the support system is to set up a dataset per player, which allows for tracking during the season and machine learning. Future research to refine the machine learning models may include the influence of contextual factors such as home or away, score, ranking, and player position.

## Figures and Tables

**Figure 1 entropy-23-00952-f001:**
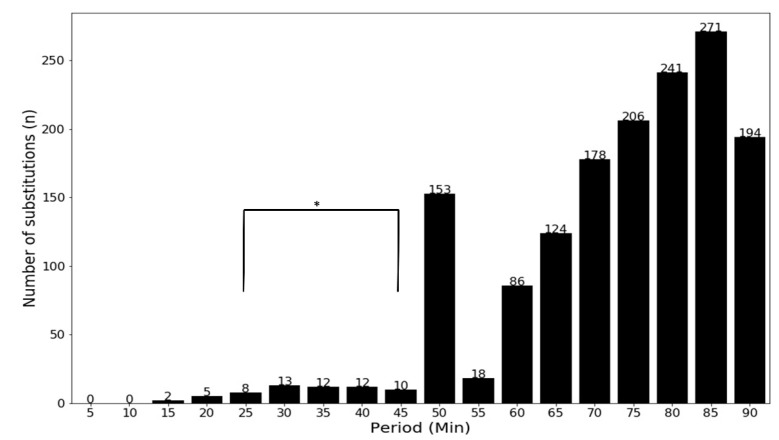
Number of substitutes per 5 min period. * Significantly lower number of substitutes (*p* < 0.001).

**Figure 2 entropy-23-00952-f002:**
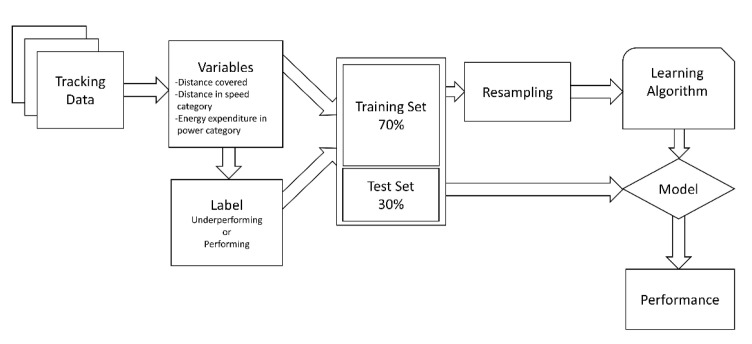
Visualization of the machine learning process.

**Figure 3 entropy-23-00952-f003:**
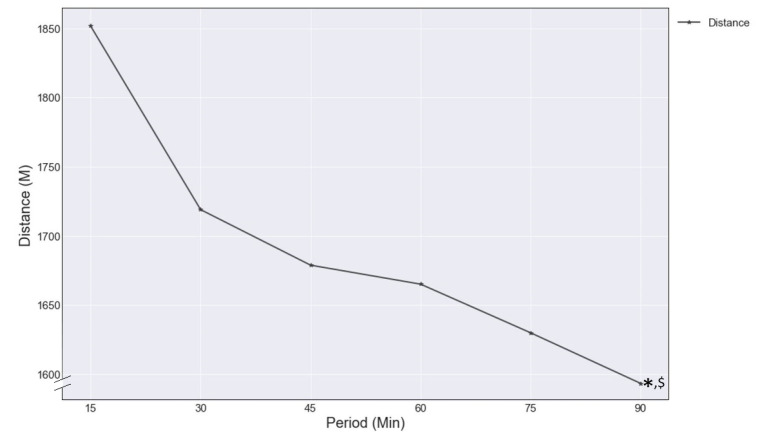
Average distance covered by entire-match players in 15 min periods. * (*p* < 0.01) a significant decline between the first half (15–45 min) and the second half (60–90 min). $ (*p* < 0.01) a significant decline between the 15-min periods in the second half (60–90 min).

**Figure 4 entropy-23-00952-f004:**
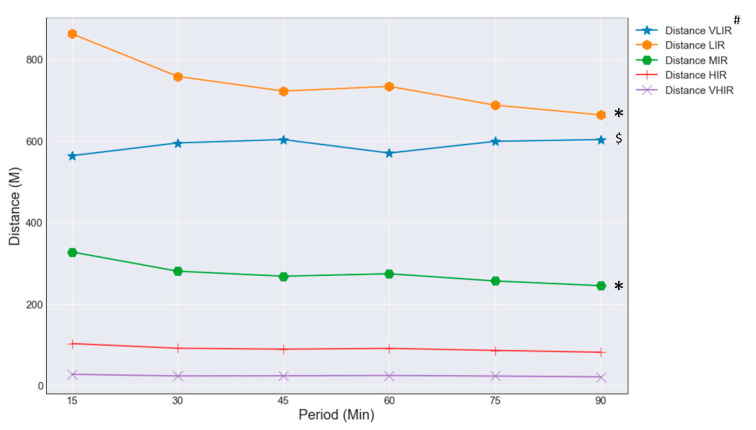
Distance covered in speed category entire match in 15 min periods. # Abbreviations of the power categories VLIR, Very Low Intensity Running; LIR, Low Intensity Running; MIR, Medium Intensity Running; HIR, High Intensity Running; VHIR, Very High Intensity Running. * (*p* < 0.01) a significant decline between the first half (15–45 min) and the second half (60–90 min). $ (*p* < 0.01) a significant increase between the 15 min periods in the second half (60–90 min).

**Figure 5 entropy-23-00952-f005:**
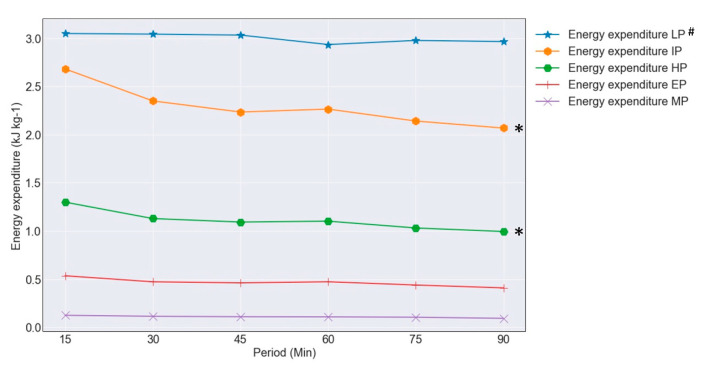
Energy expenditure in power category entire-match players in 15 min periods. # Abbreviations of the power categories: LP, Low Power; IP, Intermediate Power; HP, High Power; EP, Elevated Power; MP, Maximum Power. * (*p* < 0.01) a significant decline between the first half (15–45 min) and the second half (60–90 min).

**Figure 6 entropy-23-00952-f006:**
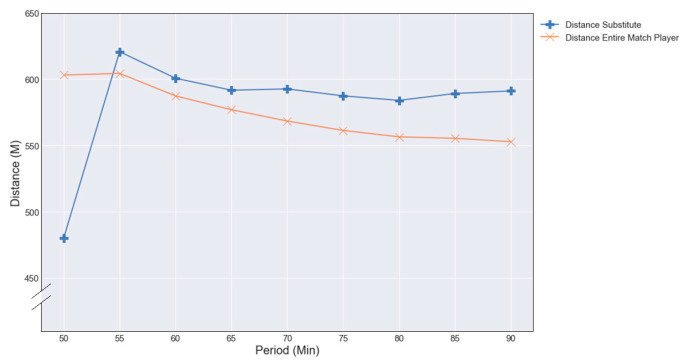
Average distance entire-match players versus substitutes second half.

**Figure 7 entropy-23-00952-f007:**
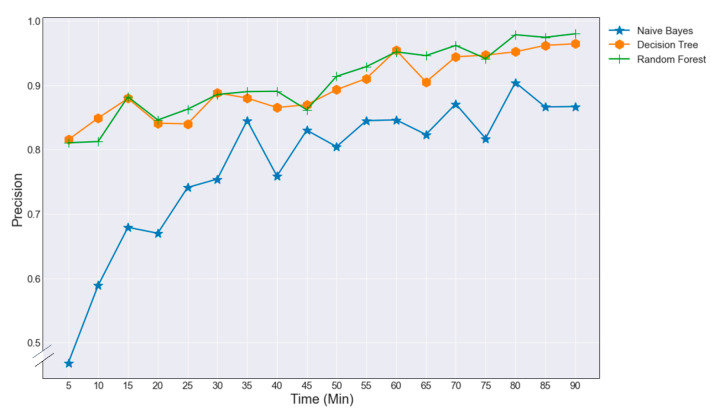
Random forest precision underperforming energy expenditure at 95% threshold in 5 min periods.

**Table 1 entropy-23-00952-t001:** Variable-based constructed features.

**Variable: Distance Covered**	
**Feature**	**Explanation**
Period	The five-minute period indicator of the match (range: 1–18)
Percentage average distance	The percentage of distance covered versus the average percentage of distance covered in the specific five-minute period
Percentage average percentage summed distance	The percentage of the summed distance covered versus the average percentage summed distance covered up to and including the specific the five-minute period
**Variable: Distance in Speed Category**	
**Feature**	**Explanation**
Period	The five-minute period indicator of the match (range: 1–18)
Percentage very low intensity running distance covered versus average percentage very low intensity running distance covered	The percentage of the distance covered in the very low intensity running speed category versus the average percentage distance in the very low intensity running speed category in the specific five-minute period.
Percentage summed very low intensity running distance covered versus average very low intensity running distance covered	The percentage of the summed distance covered in the very low intensity running speed category versus the average percentage summed distance in the very low intensity running speed category in the specific five-minute period.
Percentage low intensity running distance covered versus average percentage low intensity running distance covered	The percentage of the distance covered in the very low intensity running speed category versus the average percentage distance in the very low intensity running speed category in the specific five-minute period.
Percentage summed low intensity running distance covered versus average percentage summed Low intensity running distance covered	The percentage of the summed distance covered in the low intensity running speed category versus the average percentage summed distance covered in the low intensity running category up to and including the specific five-minute period.
Percentage medium intensity running distance covered versus average percentage medium intensity running distance covered	The percentage of the distance covered in the medium intensity running speed category versus the average percentage distance covered in the medium intensity running speed category in the specific five-minute period
Percentage summed medium intensity running distance covered versus average medium intensity running distance covered	The percentage of the summed distance covered in the medium intensity running speed category versus the average percentage summed distance covered in the medium intensity running speed category up to and including the specific five-minute period.
Percentage high intensity running distance covered versus average percentage high intensity running distance covered	The percentage of the distance covered in the high intensity running speed category versus the average percentage distance covered in the high intensity running speed category in the specific five-minute period.
Percentage summed high intensity running distance covered versus average percentage summed high intensity running distance covered	The percentage of the summed distance covered in the high intensity running speed category versus the average percentage summed distance covered in the high intensity running speed category up to and including the specific five-minute period.
Percentage very high intensity running distance covered versus average percentage very high intensity running distance covered	The percentage of the summed distance covered in the very high intensity running speed category versus the average percentage summed distance covered in the very high intensity running speed category in the specific five-minute period.
Percentage summed very high intensity running distance covered versus average percentage very high intensity running distance covered	The percentage of the summed distance covered in the very high intensity running speed category versus the average percentage summed distance covered in the very high intensity running speed category up to and including the specific five-minute period.
**Variable: Energy Expenditure in Power Category**	
**Feature**	**Explanation**
Period	The five-minute period indicator of the match (values: 1–18)
Percentage low-power energy expenditure versus average low-power energy expenditure	The percentage of the energy expenditure in the low-power category versus the average percentage energy expenditure in the low-power category in the specific five-minute period.
Percentage summed low-power energy expenditure versus average percentage summed low-power energy expenditure	The percentage of the summed energy expenditure in the low-power category versus the average percentage summed energy expenditure in the low-power category up to and including the specific five-minute period.
Percentage intermediate-power energy expenditure versus average percentage low-power energy expenditure	The percentage of the energy expenditure in the intermediate-power category versus the average percentage energy expenditure in the low-power category in the specific five-minute period.
Percentage summed intermediate-power energy expenditure versus average percentage summed intermediate-power energy expenditure	The percentage of the summed energy expenditure in the low-power category versus the average percentage summed energy expenditure in the intermediate-power category up to and including the specific five-minute period.
Percentage high power energy expenditure versus average percentage high-power energy expenditure	The percentage of the energy expenditure in the high-power category versus the average percentage energy expenditure in the high power category in the specific five minute period.
Percentage summed high-power energy expenditure versus average percentage summed high power energy expenditure	The percentage of the summed energy expenditure in the high-power category versus the average percentage summed energy expenditure in the high-power category up to and including the specific five-minute period.
Percentage elevated-power energy expenditure versus average elevated-power energy expenditure	The percentage of the energy expenditure in the elevated-power category versus the average percentage energy expenditure in the elevated-power category in the specific five-minute period.
Percentage summed elevated-power energy expenditure versus average summed elevated-power energy expenditure	The percentage of the summed energy expenditure in the elevated-power category versus the average percentage summed energy expenditure in the elevated-power category up to and including the specific five-minute period.
Percentage maximal-power energy expenditure versus average percentage maximal-power energy expenditure	The percentage of the energy expenditure in the maximal-power category versus the average percentage energy expenditure in the maximal-power category in the specific five-minute period.
Percentage summed maximal-power energy expenditure versus average percentage summed maximal-power energy expenditure	The percentage of the summed energy expenditure in the maximal-power category versus the average percentage summed energy expenditure in the maximal-power category up to and including the specific five-minute period.

**Table 2 entropy-23-00952-t002:** Variable distribution of the performing and underperforming players.

Variables	Threshold 100%	Threshold 95%	Threshold 90%
Distance Covered			
Underperforming (n)	38,490	60,820	68,347
Performing (n)	30,590	8260	733
Distance in Speed Category Model			
Underperforming (n)	42,014	64,340	69,520
Performing (n)	27,866	5540	360
Energy Expenditure in Power Category			
Underperforming (n)	34,416	7912	1604
Performing (n)	35,463	61,967	68,275

**Table 3 entropy-23-00952-t003:** Machine learning metrics.

**Variable: Distance Covered**							
**Threshold**	**Algorithm**	**Accuracy**	**AUC**		**Precision**	**Recall**	**F1-Score**
100%	Random Forest	0.90	0.94	Underperforming	0.97	0.92	0.94
				Performing	0.70	0.84	0.76
	Decision Tree	0.88	0.86	Underperforming	0.96	0.90	0.93
				Performing	0.64	0.82	0.72
	Naïve Bayes	0.57	0.58	Underperforming	0.83	0.59	0.69
				Performing	0.21	0.48	0.29
95%	Random Forest	0.75	0.79	Underperforming	0.71	0.67	0.69
				Performing	0.78	0.81	0.79
	Decision Tree	0.77	0.82	Underperforming	0.73	0.72	0.72
				Performing	0.80	0.81	0.81
	Naïve Bayes	0.73	0.75	Underperforming	0.68	0.67	0.67
				Performing	0.77	0.78	0.77
90%	Random Forest	0.93	0.95	Underperforming	0.55	0.87	0.67
				Performing	0.99	0.94	0.96
	Decision Tree	0.92	0.88	Underperforming	0.51	0.85	0.63
				Performing	0.99	0.93	0.96
	Naïve Bayes	0.74	0.74	Underperforming	0.15	0.52	0.24
				Performing	0.95	0.92	0.84
**Variable: Distance in Speed Category Model**							
**Threshold**	**Algorithm**	**Accuracy**	**AUC**		**Precision**	**Recall**	**F1-Score**
100%	Random Forest	0.89	0.96	Underperforming	0.85	0.87	0.86
				Performing	0.91	0.90	0.91
	Decision Tree	0.74	0.81	Underperforming	0.65	0.73	0.68
				Performing	0.81	0.75	0.78
	Naïve Bayes	0.70	0.78	Underperforming	0.59	0.77	0.67
				Performing	0.82	0.65	0.72
95%	Random Forest	0.96	0.98	Underperforming	0.68	0.91	0.78
				Performing	0.99	0.96	0.98
	Decision Tree	0.94	0.92	Underperforming	0.59	0.89	0.71
				Performing	0.99	0.95	0.97
	Naïve Bayes	0.97	0.83	Performing	0.23	0.72	0.35
				Performing	0.97	0.80	0.87
90%	Random Forest	1.00	0.99	Underperforming	0.61	0.94	0.74
				Performing	1.00	1.00	1.00
	Decision Tree	0.99	0.97	Underperforming	0.41	0.94	0.57
				Performing	1.00	0.99	1.00
	Naïve Bayes	0.88	0.89	Underperforming	0.03	0.74	0.06
				Performing	1.00	0.88	0.93
**Variable: Energy Expenditure in Power Category**							
**Threshold**	**Algorithm**	**Accuracy**	**AUC**		**Precision**	**Recall**	**F1-Score**
100%	Random Forest	0.89	0.96	Underperforming	0.88	0.89	0.89
				Performing	0.89	0.89	0.89
	Decision Tree	0.82	0.92	Underperforming	0.82	0.81	0.82
				Performing	0.82	0.82	0.82
	Naïve Bayes	0.81	0.90	Underperforming	0.81	0.81	0.81
				Performing	0.80	0.80	0.80
95%	Random Forest	0.97	0.99	Underperforming	0.83	0.91	0.87
				Performing	0.99	0.98	0.98
	Decision Tree	0.96	0.98	Underperforming	0.74	0.85	0.79
				Performing	0.98	0.97	0.98
	Naïve Bayes	0.90	0.87	Underperforming	0.36	0.5	0.09
				Performing	0.90	0.99	0.94
90%	Random Forest	1.00	0.99	Underperforming	0.88	0.86	0.87
				Performing	1.00	1.00	1.00
	Decision Tree	0.96	0.98	Underperforming	0.74	0.85	0.79
				Performing	0.98	0.97	0.98
	Naïve Bayes	0.99	0.51	Underperforming	0.03	0.02	0.03
				Performing	0.99	0.99	0.99

AUC = Area Under Curve.

## Data Availability

Access to the data and the definition of the database is available on Github (https://github.com/dijkhuist/Early-Performance-Prediction-Machine-Learning-in-Soccer, accessed on 24 July 2021).

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
