# Peer review of "Early Prediction of Physical Performance in Elite Soccer Matches—A Machine Learning Approach to Support Substitutions"

_entropy, 2021, doi:10.3390/e23080952_

Round 1

Reviewer 1 Report

Comments:

  • Introduction - it is quite well presented with current studies on the subject, I propose that in this chapter to introduce as clearly as possible what was the novelty of this study.
  • Keywords: after each keyword the authors must put point and comma, the authors just put only comma
  • Line 157 - The authors say: ''Variables were calculated per five-minute period of the match'', my question is what was the reasoning for which the authors calculated the variables for this time period (i.e. five-minutes), do you consider that the period is sufficient?
  • Point after each title of tables, see tables 1, 2,,3 .....and for figures.
  • For Figure 1 Visualization of the Machine Learning Process -  it is necessary for this figure to be clearer, it is not very visible on the page
  • Why was the subjects subchapter presented in the results chapter, and not in the methods chapter? Also, the authors say: ''Four hundred and eighty players participated in 302 matches. This resulted in 4935  players completing an entire match and 1522 players participating as a substitute''....my question is how was this data centralized, based on why? were there any observation or registration sheets for the participants? the method of collection is not understood.....the authors must detailed this
  • Table number 2 looks very ugly, the authors should correct it for a good understanding by the readers and in some places the abbreviations used should be written under the tables (where is the case)
  • References - are not in accordance with the journal policy, please read the instructions for the authors, chapter references

Reviewer 2 Report

ABSTRACT

Lines 18-20: just determine the substitution based on physical performance? What about technical/tactical performance or other factors that concur for overall performance?

Lines 20-21: contextualize the data collection and add the eligibility criteria.

Line 24: detail the machine learning approaches/techniques

Lines 25-25: add the values that confirm the statement

INTRODUCTION

Line 25: add some description of bioenergetics that support the statement.

Lines 26-28: specific the percentage of decline in different outcomes/ running intensities

Line 42: affected physical performance? Or overall performance?

Line 56: add examples of the devices that provide real-time motion data

Lines 58: physiological variables with real-time motion analysis? Physical?

Line 71: how can the reliability of instruments influence type 2?

Lines 73-74: support the statement with a reference

Line 82: contextual factors?

MATERIALS AND METHODS

Start the first section using an “experimental approach to the problem”. Information of study design and settings must be placed early in methods. Consider following the STROBE guidelines.

Line 100: add the eligibility criteria.

Line 109: add the reliability levels for the different outcomes analyzed in this study.

RESULTS

Line 205: add the effect size after the p-value. Same for the remaining comparisons.

Lines 219, 225, 235, 260, 263, 266, 272, 274: add the effect size after p-value.

Table 3: any chance of adding the confidence intervals for the values?

DISCUSSION

Generally good. Maybe can add a section of conclusions and add a sub-section of “practical implications” with some highlights for practitioners and eventually future research.

Round 2

Reviewer 1 Report

Lines 312- 316 - I don't think the font size is the same as the rest of the article

Reviewer 2 Report

The article was improved. Can be accepted.